# Analysis of the Impact of Media Trust on the Public’s Motivation to Receive Future Vaccinations for COVID-19 Based on Protection Motivation Theory

**DOI:** 10.3390/vaccines9121401

**Published:** 2021-11-26

**Authors:** Zeming Li, Xinying Sun

**Affiliations:** Department of Social Medicine and Health Education, School of Public Health, Peking University, Beijing 100191, China; zemingli9669@163.com

**Keywords:** COVID-19, vaccination, motivation, media trust, protection motivation theory

## Abstract

**Object**: Media trust is one of the essential factors affecting health behavior. Based on the protection motivation theory (PMT), this study explores the impact of different public media trust (traditional media, social media, interpersonal communication) on future COVID-19 vaccine motivation. **Methods**: The online survey was conducted from 14 April to 30 April 2021, and 2098 adults were recruited to participate in the online survey through the Wenjuanxing online survey platform. The survey included the PMT constructs (threat appraisal, coping appraisal, and motivation for future COVID-19 vaccination), trust in different media, vaccine hesitation reasons, and implementation of other non-pharmaceutical interventions. Structural equation model (SEM) was used for latent variable analysis, and Spearman linear correlation coefficient matrix was used to explore the relationships between variables. **Results**: In terms of trust in different media, participants who had a higher education level (*p* = 0.038), who was married (*p* = 0.002), and who had not been vaccinated against COVID-19 during the survey (*p* = 0.002) show greater trust in traditional media. Participants who were married (*p* = 0.001), who had a relatively high income (*p* = 0.020), and who had not been vaccinated (*p* = 0.044) show greater trust in social media. Older participants (*p* < 0.001) and married (*p* < 0.001) showed greater trust in interpersonal communication. In the structural equation, trust in traditional media had a direct positive impact on perceived severity (β = 0.172, *p* < 0.001) and a direct negative impact on internal rewards (β = −0.061, *p* < 0.05). Trust in both traditional and social media separately had a direct positive impact on self-efficacy (β = 0.327, *p* < 0.001; β = 0.138, *p* < 0.001) and response efficiency (β = 0.250, *p* < 0.001; β = 0.097, *p* < 0.05) and a direct negative impact on response costs (β = −0.329, *p* < 0.001; β = −0.114, *p* < 0.001). Trust in interpersonal communication had a direct positive impact on external rewards (β = 0.186, *p* < 0.001) and response costs (β = 0.091, *p* < 0.001). Overall, traditional media trust had an indirect positive influence on vaccine motivation (β = 0.311), social media trust had an indirect positive influence on vaccine motivation (β = 0.110), and interpersonal communication had an indirect negative influence on vaccine motivation (β = −0.022). **Conclusion**: This study supports the use of PMT as an intermediate variable to explore the effect of media trust on vaccination intention. High trust in traditional media has helped reduce vaccine hesitation, increased the public’s future COVID-19 vaccination motivation, and maintained other non-pharmacological interventions. Social media also had a certain promotion effect on vaccine motivation. In this context, attention should also be paid to interpersonal communication, and the science publicity work was suggested for an individual’s family members and friends in the future to improve the quality and ability of interpersonal communication.

## 1. Background

The gradual outbreak of Corona Virus Disease 2019 (starting now referred to as COVID-19) at the end of 2019 has been a global epidemic for more than a year, seriously endangering human life and health and causing significant damage to social and economic development. Global public health practices have demonstrated that vaccination has been one of the safest and most effective measures to prevent, control, and eliminate infectious diseases [1]. At present, the COVID-19 vaccine has been successfully developed and put into use in most countries. However, due to the variability of viruses such as the variant strain Delta, relevant experts said that the need for vaccine-reinforced injection should be deliberately considered in the future. Vaccine research and development with a high coverage rate is essential for establishing herd immunity and achieving immune prevention, which usually takes several years. Therefore, although the current COVID-19 vaccine has been successfully developed and put into use, the short-term development of the vaccine still makes the public delay or refuse to vaccinate. It has a specific impact on the future of public vaccination intention, ultimately influencing the establishment of the herd immune barrier [2,3,4,5].

Media is one of the most important methods for health intervention [6]. The development of media technology, diversified health information, and convenience of contact has profoundly impacted people’s cognition, attitude, and behavior concerning health. Especially for individuals during an epidemic, the public demand for information is higher. It is necessary to collect relevant information through various channels, reduce the uncertainty of threats, form the subjective cognition of risks, and guide individual behaviors. Different media have different effects on information transmission [7]. For example, traditional media such as TV news and newspapers emphasize the connection with sources of expert information [8]. In the face of a social crisis, people rely more on mass media to obtain authoritative information, transmit disease knowledge, and improve public risk awareness [9,10]. They also tend to seek the help of social media to obtain information and exchange opinions [11]. In addition, media information has helped drive negative press and ultimately led to vaccine hesitation [12].

At the same time, media trust is one of the essential factors affecting health behavior, which has a certain influential role in health communication. People are often highly dependent on mass media when perceiving a threat in a social crisis [8,13]. With the development of digital society, social media plays an increasingly important role in disseminating health or environmental information during the public health crisis [14]. For instance, people with high trust in mass media are more likely to accept health information and adopt recommendation behavior [15,16]. High trust in social media is positively associated with personal protective behavioral intentions [17]. There is also a positive correlation between media trust and its usage [18,19], and the Rational Audience model believes that the audience will focus more on the medium they trust [18].

Meanwhile, media trust plays a vital role in influencing individual acceptance of health information conveyed in the public crisis [20], thus affecting the ultimate behavioral intention [21]. In addition, relevant studies have shown that media trust often has cognitive and emotional effects on recipients [22]. For example, high media trust often leads to increased concern about influenza [23], while low media trust causes uncertainty to increase emotional responses such as fear and worry [24]. Previous studies have found that behavior theories can explain behavior from social and mental health perspectives and help medical and health care providers design effective interventions to alter specific behaviors [25,26].

Protection Motivation Theory (PMT) has been one of the most commonly used theories to explain how individuals take health-promoting measures. It is an important theory to explore and explain behavior formation from social and psychological perspectives, which has been widely applied and confirmed in many fields such as smoking, chronic diseases, infectious diseases, vaccination, etc. [27,28,29]. PMT considers that people will first conduct a threat appraisal pathway and a coping appraisal pathway when faced with a threat and adopt protective behaviors, where the threat appraisal takes into account the severity of the threat and the perceived susceptibility to the threat, which includes four factors:Severity, which assesses the perceived negative consequences from a risk behavior.Vulnerability, which assesses the perceived likelihood of the individual being affected by Potential negative consequences.Intrinsic rewards, which assess the perceived positive physical and psychological effect from engaging in a risk behavior.Extrinsic rewards, which assess the perceived positive social reactions or consequences of engaging in the risk behavior.

The coping appraisal evaluates whether the recommended behavior effectively mitigates the threat and whether they can carry out the behavior and includes three factors:Self-efficacy, which perceived ability to adopt a protective behavior.Response efficacy, which assessed the effectiveness of the protective behavior in lessening the health threat.Response costs consist of the perceived social, monetary, personal, time and effort costs from adapting the protective.

Previous studies have also shown that threat beliefs posed by diseases and attitudes towards performing recommended behaviors are essential in interpreting various protective behaviors [30,31]. For instance, perceived severity, susceptibility, response efficacy, self-efficacy, and response cost from PMT were highly explanatory of vaccination behavior during the influenza pandemic [25], and higher perceived severity and self-efficacy can predict more of home isolation behavior [32] and intention [33]. Therefore, the explanation of the PMT on the public’s future COVID-19 vaccine motivation from social and psychological perspectives is worth further investigation.

Based on the PMT, several studies have analyzed the impact of various media-related factors on protection behavior during the pandemic, including the impact of Internet resources on self-isolation [32], the impact of information tips on self-isolation [34], and the impact of information sources on vaccination intention [35]. However, in the new context of COVID-19, the existing literature has not elaborated on the impact of media trust on the motivation of future vaccine behavior. Media trust is one of the essential factors affecting health behavior, which is most likely to influence general cognition and emotion, ultimately affecting behavior intention. Especially under the situation that there is some public hesitation about the COVID-19 vaccine for various reasons or conditions, different media trust would have a certain impact on the future vaccination intention. Therefore, based on the PMT, this study explores the impact of different public media trust (traditional media, social media, interpersonal communication) on future COVID-19 vaccine motivation. The results will help decision-makers provide scientific, appropriate health transmission and intervention strategies for future disease pandemics.

## 2. Materials and Methods

### 2.1. Participants and Procedures

According to the age stratification of China’s sixth national population census at 18–30 years old (20.32%), 30–50 years old (33.43%), 50–70 years old (19.49%) and the gender ratio (1:1), this study recruited adults over the age of 18 to participate in an online survey through the platform Wen Juan Xing during 14 April to 30 April 2021. As structural equation modelling guarantees a sample size to observed variable ratio observed Variable of at least 10:1, with 54 observed variables in this study, a minimum of 540 subjects would be required. Participants filled in the questionnaire with a smartphone or computer through the link released by the platform and only completed it once with each ID number. Those who handed in a valid questionnaire were given a gift for acknowledgment. Quality control questions were set in the questionnaire, and questionnaires with illogical answers or the same answers for all options were treated as invalid. A total of 2098 valid questionnaires were collected after arrangement. The project was reviewed and approved by the Biomedical Ethics Committee of Peking University (IRB00001052-20081), and all the respondents were aware of the project content and signed the informed consent.

### 2.2. Measures

#### 2.2.1. Measures of Protection Motivation Theory

Measures for the protective motivation theory used in this study are listed in Table 1.

#### 2.2.2. Media Trust for COVID-19

This study divided media into three categories: traditional media, social media, and interpersonal communication. Each was scored using a Likert Scale of 1 (very low) to 5 (very high). The higher the score for each item, the higher the trust in the medium.

#### Trust in Traditional Media

Trust in traditional media was measured by asking participants the degree of trust in the epidemic-related information published by the following different traditional media in the past month, including (i) national and local TV news, broadcasts, newspapers; (ii) interview with medical experts; (iii) government website.

#### Trust in Social Media

Trust in social media was measured by asking participants the degree of trust in the epidemic-related information published by the following different social media in the past month, including (i) WeChat and Weibo official accounts of national and local official media; (ii) TikTok, Weibo and WeChat official accounts of influential self-media operators; (iii) medical science platforms such as DingXiangYiSheng.

#### Trust in Interpersonal Communication

Trust in interpersonal communication was measured by asking participants the degree of trust in the “information released by WeChat groups or WeChat moment in the past month” and the “epidemic-related information obtained from conversations with relatives, friends and neighbors”.

#### 2.2.3. Other Related Information of COVID-19

##### External Reasons for Vaccine Hesitation

Participants were asked about the reasons for hesitation before or without COVID-19 vaccination, and the reasons were classified into four categories with a total of three points of three items, namely (i) the existing non-pharmacological intervention (NPI) is enough for protection; (ii) the vaccine is unsafe and has side effects; (iii) the appointment process is complicated; (iv) the personal time and physical conditions are limited, with 1 for “Yes” and 0 for “No”.

The reasons why participants were hesitant about the COVID-19 vaccine were divided into four categories, with each category consisting of three items, 12 points in total, namely:(i).the existing Non-Pharmacological Intervention (NPI) is enough for protection;(ii).Safety and possible side effects;(iii).Complex appointment process;(iv).Limited by personal conditions (time and physical conditions),

Response options to both questions were yes (1) or no (0).

##### Implementation of Other Protection Behaviors

To assess how often participants washed hands in the past month, we used three items: “wash your hands before eating any food”, “the first thing is to wash and disinfect your hands when getting home from the outside”, and “everything (take-out, express, supermarket goods) you bring home from the outside should be disinfected and cleaned”. To assess how often participants wore a mask in the past month, we used two items: “wear a mask when going out” and “wear a mask in a confined space or public place”. The Likert Scale was used for scoring, from 1 (seldom) to 5 (every day). The higher the score for each item, the better the performance of the behavior.

##### Socio-Demographic Questions

This study also asked socio-demographic questions about gender, age, educational level, marriage, income, vaccination, etc.

### 2.3. Data Analysis

This study used SPSS 26.0 for statistical analysis and Mplus8.0 for constructing the structural equation model. Frequency, percentage, mean and standard deviation were used to describe participant characteristics, and variable scores, t-test, one-way ANOVA, or Kruskal–Wallis H test were used for monofactor analysis. Structural equation model (SEM) was used for latent variable analysis, and Spearman linear correlation coefficient matrix was used to explore the relationships between variables, and the Maximum Likelihood estimation method was used for parameter estimation. SEM is a multivariate statistical method for modeling relationships utilizing covariance of variables. Generally speaking, SEM can be decomposed into two parts: measurement model and structural model. The former refers to the relationship between indicators and latent variables and deals mainly with the measurement of latent variables; the latter deals with the relationship between latent variables and with observed variables other than indicators measured by non-latent variables and deals mainly with the causal relationships between different concepts. Hence SEM is often used to test the relationships between theoretical hypotheses. In this study, we will use SEM to explore the relationship between different channels of media trust (observed variables) and each of the latent variables of PMT and assesses the impact on vaccination intention. The SEM was evaluated according to relevant indicators: Turker–Lewis Index (TLI) and Comparative Fit Index (CFI). Usually higher than 0.95 indicates that the model fits well, higher than 0.90 are regarded as the accepted standard [36]. Standardized Root Mean Square Residual (SRMR), When the value is less than 0.08, it indicates that the model fits well [36], and when the value of Root Mean Square Error of approximation (RMSEA) is less than 0.05, the model fits well [37]. Test level α = 0.05.

## 3. Results

The average age of the 2098 participants was 31.22 ± 8.29, the number of males was 1114, and the male-female ratio was 1.13:1. The level of education was mainly undergraduate, the marital status was mainly married, and the per-capita monthly household income was 5000–9999 yuan. In the past COVID-19 vaccination behavior, vaccinated participants accounted for 40.99% during the survey (Table 2).

In terms of trust in different media, participants who had a higher education level (*p* = 0.038), who were married (*p* = 0.002), and who had not been vaccinated against COVID-19 during the survey (*p* = 0.002) show greater trust in traditional media. Participants who were married (*p* = 0.001), who had a relatively high income (*p* = 0.020), and who had not been vaccinated (*p* = 0.044) show greater trust in social media. Older participants (*p* < 0.001) and married (*p* < 0.001) showed greater trust in interpersonal communication, as shown in Table 3.

The correlations between various variables are shown in Table 4. (i) Different types of media trust had specific effects on public vaccine hesitancy. Trust in both traditional and social media was negatively correlated with the causes of vaccine hesitation that “advocation for NPI” (r = −0.127, *p* < 0.001; r = −0.088, *p* < 0.001) and “worrying about safety and possible side effects” (r = −0.093, *p* < 0.001; r = −0.074, *p* < 0.001). Trust in traditional media was positively correlated with “the complex appointment process” in vaccine hesitation (r = 0.068, *p* < 0.001). (ii) Three types of media trust show a positive correlation with future vaccination motivation. (iii) Trust in traditional and social media tended to coincide with perceived severity, intrinsic rewards, extrinsic rewards, response efficacy, response costs, and self-efficacy of the PMT, but trust in interpersonal communication was positively correlated with extrinsic rewards (r = 0.059, *p* < 0.001). (iv) The “advocation for NPI” (r = −0.185, *p* < 0.001) and the “worrying about safety and possible side effects” (r = −0.143, *p* < 0.001) in the cause of vaccine hesitation were negatively correlated with vaccination motivation. The reason for vaccination hesitation, which is the “complicated appointment process” was positively correlated with the vaccination motivation (r = 0.215, *p* < 0.001).

As shown in Table 4, the structural equation model was constructed according to the Pearson correlation coefficient matrix of each variable and the fitting information (Figure 1), and the fitting indexes were as follows: X2/df = 3.32, CFI = 0.938, TLI = 0.929, RMSEA = 0.033, SRMR = 0.040, which indicated that the model was reasonably fit. In the structural equation, Trust in traditional media had a direct positive impact on perceived severity (β = 0.172, *p* < 0.001) and a direct negative impact on internal rewards (β = −0.061, *p* < 0.05). Trust in both traditional and social media separately had a direct positive impact on self-efficacy (β = 0.327, *p* < 0.001; β = 0.138, *p* < 0.001) and response efficiency (β = 0.250, *p* < 0.001; β = 0.097, *p* < 0.05) and a direct negative impact on response costs (β = −0.329, *p* < 0.001; β = −0.114, *p* < 0.001). Trust in interpersonal communication had a direct positive impact on external rewards (β = 0.186, *p* < 0.001) and response costs (β = 0.091, *p* < 0.001). Overall, traditional media trust had an indirect positive influence on vaccine motivation (β = 0.311), social media trust had an indirect positive influence on vaccine motivation (β = 0.110), and interpersonal communication had an indirect negative influence on vaccine motivation (β = −0.022).

Traditional media trust had a direct negative impact on both the “advocation for NPI” (β = −0.120, *p* < 0.001) and the “worrying about safety and possible side effects” (β = 0.099, *p* < 0.001) in vaccine hesitation and had a direct positive impact on the “limited by personal conditions” (β = 0.070, *p* < 0.05). Under the influence of reasons for vaccine hesitation, traditional media trust had an indirect positive impact on vaccine motivation (β = 0.333).

In addition, traditional media trust had a direct positive impact on the behavior of “wearing a mask” (β = 0.106, *p* < 0.001), and social media trust had a direct positive impact on the behavior of “hand-washing” (β = 0.119, *p* < 0.001), and “Frequent hand-washing” had a direct negative impact on vaccine motivation (β = 0.060, *p* < 0.05).

## 4. Discussion

In April 2021, during the investigation conducted in this study, the epidemic had been under control in China. Only a few local cases had been detected in each region, but the outbreak overseas was intensifying, given that the novel variant strain Delta was ravaging in many countries such as India and the United States. To this end, the Chinese government actively advocated mass vaccination, but the public still had greater autonomy of vaccination or not as there was no recurrence of the epidemic in China at that time. Therefore, this study was more objective to analyze the impact of different media trust on the future vaccine motivation through avoiding the repeated outbreak or the government requirements and other special situations.

### 4.1. Impact of PMT on Future Vaccination Intentions

This study found a higher degree of interpretation on the public’s future vaccination intentions using PMT. It can be observed that self-efficacy and response efficacy were direct predictors of vaccine motivation, indicating that people had the confidence to solve or exclude internal or external factors affecting vaccination against COVID-19. At the same time, they strongly believed in vaccination, which can increase protection for vaccination measures in the post-pandemic period. Besides, self-efficacy present positive effects on behavioral motivation through response efficacy, consistent with the findings of Krieger [38] and Liu [27] that response efficacy was an essential mediator of self-efficacy and behavioral intention. Although people had confidence in eliminating difficulties (i.e., self-efficacy), they firmly believed that the effectiveness of specific behaviors was more important (i.e., response efficacy).

In the threat appraisal of PMT, intrinsic and extrinsic rewards harmed behavioral intention. Rogers believed that factors enhancing maladaptive behavior, such as intrinsic rewards (self-pleasure, satisfaction) and extrinsic rewards (social recognition) could reduce behavioral response probability [39]. The COVID-19 vaccine was different from other long-developed vaccines, and it took a relatively short time for research and experiment to come into use. Although national governments encouraged public vaccination, individuals continue to have concerns and misgivings about the COVID-19 vaccine to aggravate maladaptive behavior (non-vaccination) and reduce the future COVID-19 vaccine intention. In theory, both response costs and intrinsic rewards directly negatively affect behavioral motivation in PMT. However, in this study, response costs through intrinsic rewards harmed behavioral motivation, which refers to “side effects, psychological burden, increased infection rate, spending time and energy.” Response costs had no direct impact on behavioral motivation but will increase self-pleasure and satisfaction caused by non-vaccination to reduce the future motivation of vaccination.

### 4.2. Role of Media Trust of COVID-19 Vaccination

There is a significant relationship between public trust in the mass media and health behaviors [16,40]. Generally, people with high trust in the mass media influence health behaviors by increasing media usage to obtain more information or their trust in public health information [41,42,43]. However, few studies have further investigated the causal mechanisms between media trust and health behaviors. This study used PMT as an intermediate variable to further explore how media trust influences future vaccination intentions. It was found that traditional media trust influenced PMT threat appraisal (severity, vulnerability, intrinsic rewards, extrinsic rewards) and coping appraisal (response efficacy, self-efficacy, response cost), but social media trust only influenced the coping appraisal.

In this study, high trust in traditional media can significantly improve the public’s self-efficacy and response efficacy and reduce intrinsic rewards, extrinsic rewards, and response costs of maladaptive response to improve the public’s future vaccine motivation, which indicated that during the public health crisis, China’s traditional media represented by national and local TV news, broadcasts, newspapers, interviews with medical experts, and official websites of government departments were still the most traditional media, which were usually in line with the trend of national policies. During the COVID-19 vaccination period, the Chinese government provided free COVID-19 vaccination services to the general public, advocating that the public receive the COVID-19 vaccine “as much as possible if they should” to improve herd immunity. Government media such as national and local television news and newspapers promoted vaccination regarding indications, safety, vaccination procedures, and free vaccination. At the same time, medical experts publicized it from newly confirmed cases, mass vaccination, virus variability, and vaccine protection effect, so that people with a high level of trust in traditional media not only had less reason to hesitate about the vaccine (support for NPI and worrying about safety and efficacy) but also were motivated to take the protective measures recommended by the government. The findings of this study also differ from those of Nioi [12] and Zhao [44] et al. In foreign countries, different types of coverage of the same issue may occur due to the different political orientations of media outlets. Therefore, public trust in different media often leads to behavioral differences such as reduced protective behavior or increased vaccine hesitancy [45]. In China, however, both national and local traditional media outlets tend to have the same political orientation, i.e., advocating for the public to be vaccinated as much as possible. Therefore, a high level of trust in traditional media helps to reduce vaccine hesitancy and increase future vaccination intentions.

Moreover, people with greater trust in traditional media continue to recognize the seriousness of COVID-19 for humans even in the later stages of the epidemic and thus help maintain protective behaviors of ‘wearing a mask.’ The higher the perceived severity of COVID-19, the more motivated the public to maintain non-pharmacological protective behaviors due to past protective experiences. In a word, when individuals perceive COVID-19 as a severe threat to their health, they are highly likely to take protective measures to prevent disease [34], but the prevailing environment and past experiences may influence behavioral decisions. From the perspective of public health, more trust in traditional media can increase individuals’ acceptance of such information to a greater extent. Thus, they can always agree that infection with COVID-19 will severely impact the human body, psychology, income, and life and take safety measures recommended by the government.

In this study, social media trust had no effect on perceptions of COVID-19 severity and susceptibility, which is consistent with Ranjit’s findings in the USA [34]. Although social media trust also directly impacted self-efficacy, response efficacy, and response costs, the degree was slightly lower than traditional media trust, related to social media participation and information diversification [46]. Reviews and widespread discussion of COVID-19 can still be found on social media such as Weibo, and misinformation or disinformation, including “conspiracy” may affect the proper understanding of the health, social and economic threats posed by COVID-19 [47] and may reduce political support [48]. As a result, individuals with high trust in social media did not have much motivation to comply with measures recommended by the government. On the other hand, there was a direct positive impact between social media trust and “frequent hand-washing,” but it will reduce the future vaccine motivation when people believe that hand-washing and other NPI measures were enough to protect them from the virus. In general, individuals with higher trust in social media would receive false information such as exaggerated adverse vaccine effects, low safety, and non-protection effect, which influenced people to correctly understand the role and social effect of the COVID-19 vaccine, thus leading people to ignore the reality and reduce the future vaccination intention. A relevant study had shown that reducing harmful advice by only 10% can significantly curb risk-taking behaviors [49], so addressing the theory of conspiracy may be as important as disseminating health advice regarding public health, given the scope, speed, and scale of social media.

In this study, a higher degree of trust in interpersonal communication positively impacted the response costs, intrinsic rewards, and extrinsic rewards of vaccination behavior, ultimately reducing the future vaccination motivation. Previous studies have shown that interpersonal communication with family members and friends will affect health behavior and intentions, such as vaccination [50,51,52,53]. Interpersonal communication is usually referred to (direct or indirect) information communication among people. In this study, the effect of interpersonal communication trust on future vaccination motivation was contrary to traditional and social media. In the context of COVID-19, the “short-term” development and use of the COVID-19 vaccine still inspired a great debate in public. Consider the following factors: (i) The information in person-to-person communication was often poorly reviewed, making it easy to spread misinformation or rumors. (ii) People usually focused on specific information [54] based on personality, occupation, or preference and ignored other helpful information, such as only paying attention to the side effects of the COVID-19 vaccine while ignoring information related to vaccine safety. (iii) It was found in this study that compared with the younger generation, older people had more trust in interpersonal communication, who could not often identify actual or false information and believed misinformation easily. Therefore, a higher degree of person-to-person communication will magnify the cost of COVID-19 vaccination, increase the inherent pleasure and social recognition brought by non-vaccination, and reduce the future vaccination intention.

At the occurrence of a public health crisis, traditional media, social media, or interpersonal communication for spreading information was essential, which can help the public grasp the dynamics of events and kept calm [10]. In this study, compared with social media and interpersonal communication, public trust in traditional media had a stronger positive influence on future vaccination motivation and non-pharmacological interventions. In China, national and local TV news, broadcasts, newspapers, interviews with medical experts, and official websites of government departments were considered highly credible [55]. They often represented the official attitude and authority of the Chinese government, and these media had access to the most important, reliable sources of information. Therefore, high trust in traditional media can enable individuals to increase the acceptance of such information, thus producing health-related decisions and behaviors [56], such as improving the public’s future vaccination intention.

### 4.3. Limitations

This study had the following limitations. Firstly, the study was a cross-sectional study, and the relationships between variables in the model cannot be interpreted as causal; future time series analysis could be conducted to model causality. Secondly, public threat or response assessments of vaccination and future vaccination motivation changed over time, so the cross-sectional study design of this study also limited the possibility of tracking such changes. Thirdly, using an online survey may limit the representativeness of the results, so this study used a large sample size and random stratified sampling to address this limitation. In addition, this study published the questionnaire in an electronic format, thus excluding those who could not use a smartphone to answer the questionnaire, particularly those who were older, and therefore the study lacked representation of those aged 50 and over.

## 5. Conclusions

COVID-19 has been one of the greatest threats to human health in this century. In the late stage of the pandemic, national governments, experts, and scholars have suggested that the public should receive the COVID-19 vaccine as soon as possible to build herd immunity and reduce viral transmission. Meanwhile, with the emergence of variant strains, the possibility of reinforcing vaccination or a “new” vaccine should be considered in the future. However, these suggestions were accepted differently by the Chinese population. Therefore, this study explored the impact of public trust in different media on COVID-19 future vaccination motivation in this context. In China, traditional media was still the primary information transmission channel in the COVID-19 pandemic, and the information published in the later period of COVID-19 was consistent with the national policy. High trust in traditional media helped improve vaccination self-efficacy and response efficacy, reduced response costs, intrinsic rewards, extrinsic rewards, and vaccine hesitation, increased the public’s future vaccination motivation, and maintained other non-drug interventions. Although the trust in social media was less effective than traditional media, it still impacted vaccine coping appraisal and contributed to future vaccination intentions. In this context, attention should also be paid to interpersonal communication, and the science publicity work was suggested for an individual’s family members and friends in the future to improve the quality and ability of interpersonal communication.

## Figures and Tables

**Figure 1 vaccines-09-01401-f001:**
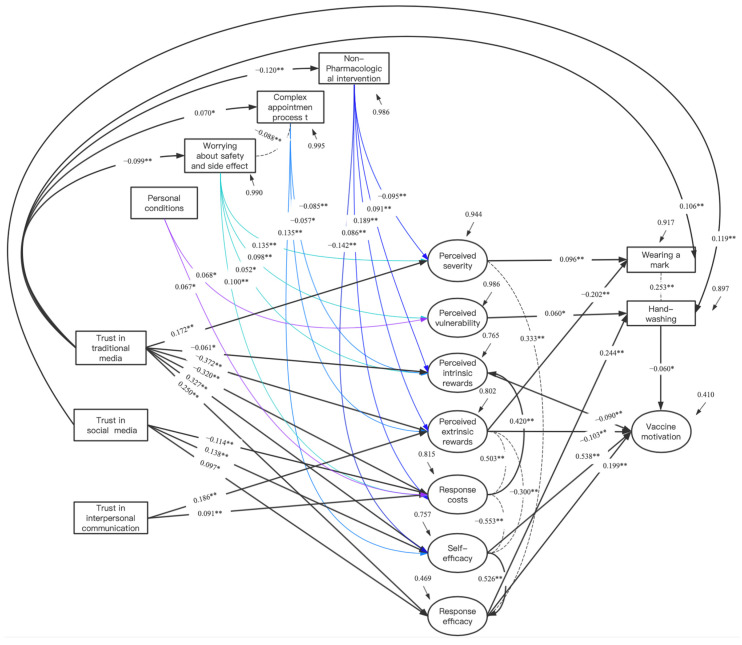
Structural equation modeling of different media trust and PMT variables. ** *p* < 0.001,* *p* < 0.05.

**Table 1 vaccines-09-01401-t001:** Measures used in the present study.

Measures	Items	Response Scale
Severity	Item 1: Infection with COVID-19 would cause serious health problems.	1 (strongly disagree) to 5 (strongly agree)
	Item 2: Infection with COVID-19 would have a detrimental effect on mental health, leading to anxiety, fear, depression, and other negative emotions.	
	Item 4: Infection with COVID-19 would have a severe impact on daily life.	
	Item 5: Infection with COVID-19 would affect one’s or a family’s financial income.	
Vulnerability	Item 1: Possibility of acquiring COVID-19 when studying and working in the same space as an infected person.	1 (not at all) to 5 (certain)
	Item 2: Possibility of acquiring COVID-19 if you use the same indoor air purification system as an infected person.	
Intrinsic rewards	Item 1: Not getting a COVID-19 jab can save people from possible adverse physical reactions.	1 (strongly disagree) to 5 (strongly agree)
	Item 2: No skin pain (from the injection) without getting COVID-19 jab	
	Item 3: No side effects to worry about without getting COVID-19 jab.	
	Item 4: Not getting a COVID-19 jab can save time and energy (no appointments, no queues).	
Extrinsic rewards	Item 1: Non-vaccination can prove good health (or stronger immune system).	1 (strongly disagree) to 5 (strongly agree)
	Item 2: Friends around me do not get vaccinated, and it would feel unusual for me to do so.	
	Item 3: Non-vaccination can prove to others that I am brave.	
	Item 4: Non-vaccination will show that I am sensible and not a follower.	
	Item 5: Non-vaccination would show that I am reasonably knowledgeable (e.g., I know that the virus will mutate).	
Self-efficacy	Item 1: I will be vaccinated even if I test negative for COVID-19.	1 (strongly disagree) to 5 (strongly agree)
	Item 2: I will be vaccinated even if there are no new confirmed cases in my city.	
	Item 3: I will get vaccinated even if people around me think it is unnecessary.	
	Item 4: I will get vaccinated even if the vaccination facility is far from me.	
	Item 5: I will get vaccinated even if I am busy with school or work.	
Response efficacy	Item 1: Vaccination is a very effective way to protect me against COVID-19.	1 (strongly disagree) to 5 (strongly agree)
	Item 2: Vaccination greatly reduces the risk of infection to my family and others around me.	
	Item 3: Vaccination helps me to concentrate more on my studies, work, and life.	
	Item 4: Vaccination helps to end the outbreak as soon as possible.	
Response costs	Item 1: Safety and possible side effects of vaccine.	1 (strongly disagree) to 5 (strongly agree)
	Item 2: Vaccination can be psychologically taxing.	
	Item 3: Vaccine is also a virus will increase the risk of infection.	
	Item 4: Vaccination will take time and effort.	
Vaccine motivation	Item 1: If a booster of COVID-19 vaccine is required in the future, will you get vaccinated?	1 (never) to 5 (certain)
	Item 2: Would you get the “new” vaccine in the future if the virus mutates and government policy recommends it?	

**Table 2 vaccines-09-01401-t002:** Characteristics of the participants (*N* = 2098).

Characteristics		*N*	%
Gender	male	1114	53.10
	female	984	46.90
Age group	18–29	928	44.23
	30–39	862	41.09
	40–49	238	11.34
	≥50	70	3.34
Education level	High School and below	180	8.58
	Undergraduate	1738	82.84
	Postgraduate and above	180	8.58
Marital status	unmarried	799	38.08
	married	1299	61.92
Monthly household income (RMB)	<2999	154	7.34
	3000–4999	370	17.64
	5000–9999	678	32.32
	10,000–14,999	407	19.40
	≥15,000	489	23.31
Past behavior	vaccinated	860	40.99
	unvaccinated	1238	59.01

**Table 3 vaccines-09-01401-t003:** A monofactor analysis of media trust in sociodemography.

Characteristics		Trust in Traditional Media	Trust in Social Media	Trust in Interpersonal Communication
Gender	male	4.08 ± 0.49	3.74 ± 0.53	2.89 ± 0.84
	female	4.08 ± 0.49	3.71 ± 0.54	2.86 ± 0.86
	t/H	−0.572	−1.586	−0.630
	*p*	0.567	0.113	0.529
Age group	18–29	4.07 ± 0.47	3.70 ± 0.51	2.78 ± 0.86
	30–39	4.10 ± 0.49	3.75 ± 0.55	2.92 ± 0.85
	40–49	4.04 ± 0.55	3.71 ± 0.57	3.05 ± 0.80
	≥50	4.10 ± 0.48	3.76 ± 0.51	3.16 ± 0.70
	F/H	5.079	6.116	34.354
	*p*	0.166	0.106	0.000
Education level	High School and below	4.00 ± 0.53	3.66 ± 0.58	2.94 ± 0.83
	Undergraduate	4.09 ± 0.49	3.73 ± 0.53	2.88 ± 0.85
	Postgraduate and above	4.06 ± 0.47	3.71 ± 0.52	2.77 ± 0.91
	F/H	6.052	2.678	4.741
	*p*	0.048	0.262	0.093
Marital status	unmarried	4.04 ± 0.51	3.68 ± 0.53	2.76 ± 0.87
	married	4.11 ± 0.48	3.75 ± 0.54	2.95 ± 0.84
	t/H	3.140	3.366	4.765
	*p*	0.002	0.001	0.000
Monthly household income (RMB)	<2999	4.05 ± 0.46	3.69 ± 0.51	2.81 ± 0.83
	3000–4999	4.04 ± 0.52	3.68 ± 0.53	2.84 ± 0.85
	5000–9999	4.09 ± 0.47	3.70 ± 0.53	2.84 ± 0.86
	10,000–14,999	4.10 ± 0.49	3.78 ± 0.54	2.93 ± 0.88
	≥15,000	4.09 ± 0.50	3.75 ± 0.55	2.93 ± 0.83
	F/H	5.072	11.635	7.718
	*p*	0.280	0.020	0.102
Past behavior	vaccinated	4.05 ± 0.51	3.70 ± 0.55	2.86 ± 0.86
	unvaccinated	4.12 ± 0.47	3.75 ± 0.53	2.91 ± 0.85
	t/H	9.382	4.074	2.240
	*p*	0.002	0.044	0.135
Total	-	4.08 ± 0.49	3.72 ± 0.54	2.88 ± 0.85

**Table 4 vaccines-09-01401-t004:** Pearson correlation in the associations between studied variables.

	1	2	3	4	5	6	7	8	9	10	11	12	13	14	15	16	17
1. Trust in traditional media	1																
2. Trust in social media	0.614 **	1															
3. Trust in interpersonal communication	0.253 **	0.394 **	1														
4. Advocation for NPI	−0.127 **	−0.088 **	−0.043	1													
5. Safety and side effects	−0.093 **	−0.074 **	−0.028	0.002	1												
6. Complex appointment process	0.068 **	0.043 *	−0.001	−0.156 **	−0.325 **	1											
7. Personal conditions	−0.003	−0.007	0.011	0.004	−0.137 **	−0.013	1										
8. Severity	0.163 **	0.097 **	0.021	−0.075 **	0.118 **	−0.005	0.015	1									
9. Vulnerability	0.026	−0.002	−0.004	−0.002	0.092 **	−0.012	0.051 *	0.208 **	1								
10. Intrinsic rewards	−0.212 **	−0.134 **	−0.002	0.178 **	0.124 **	−0.126 **	0.032	−0.027	0.051 *	1							
11. Extrinsic rewards	−0.275 **	−0.171 **	0.059 **	0.179 **	0.062 **	−0.100 **	0.004	−0.239 **	−0.027	0.300 **	1						
12. Response efficacy	0.373 **	0.325 **	0.145 **	−0.096 **	−0.056 *	0.072 **	0.007	0.255 **	0.036	−0.177 **	−0.320 **	1					
13. Response costs	−0.344 **	−0.274 **	−0.065 **	0.124 **	0.115 **	−0.070 **	0.079 **	−0.093 **	0.022	0.386 **	0.382 **	−0.371 **	1				
14. Self-efficacy	0.413 **	0.349 **	0.149 **	−0.208 **	−0.100 **	0.148 **	−0.046 *	0.156 **	0.027	−0.369 **	−0.342 **	0.496 **	−0.537 **	1			
15. Vaccine motivation	0.341 **	0.261 **	0.063 **	−0.185 **	−0.143 **	0.215 **	−0.027	0.123 **	−0.022	−0.315 **	−0.336 **	0.391 **	−0.381 **	0.552 **	1		
16. Hand-washing	0.236 **	0.231 **	0.152 **	−0.084 **	0.019	−0.037	−0.012	0.107 **	0.072 **	−0.106 **	−0.048 *	0.266 **	−0.160 **	0.259 **	0.138 **	1	
17. Wearing a mask	0.200 **	0.136 **	0.003	−0.085 **	0.042	−0.031	−0.009	0.175 **	0.023	−0.109 **	−0.217 **	0.224 **	−0.168 **	0.217 **	0.171 **	0.282 **	1

** *p* < 0.001,* *p* < 0.05.

## Data Availability

The data will be available upon reasonable request to the corresponding authors.

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
