# Peer review of "Analysis of the Impact of Media Trust on the Public’s Motivation to Receive Future Vaccinations for COVID-19 Based on Protection Motivation Theory"

_vaccines, 2021, doi:10.3390/vaccines9121401_

Round 1

Reviewer 1 Report

I read with great interest the article entitled “Analysis of the impact of media trust on the public’s motivation to receive future vaccinations for COVID-19 based on protection motivation theory”.

In their work the authors explore the impact of different public media trust (traditional media, social media, interpersonal communication) on future COVID-19 vaccine motivation.

The article is well written but has critical issues that must be corrected to make it publishable.

Major concerns.

Introduction:

Authors should acknowledge that media information has helped fuel the emergence of conspiracy theories as well as providing correct information. The administration of conflicting information has contributed to creating hesitations in a certain segment of the population. [cfr The Waiver of Patent Protections for COVID-19 Vaccines during the ongoing Pandemic and the Conspiracy Theories: Lights and Shadows of an Issue on the Ground. Front. Med. 8:756623. doi: 10.3389/fmed.2021.756623].

Another aspect that needs to be clarified is that of the fear of COVID-19. According to the authors, this aspect (considering the data of China better than other states for incidence and mortality) may have influenced the opinion of the interviewees? [cf 10.22514 / sv.2021.098].

 Methods:

1) The authors should specify how the calculation of the sample size was made.

Discussion:

In the discussion, the authors could evaluate whether the same type of study was carried out in other countries by comparing the results in a few lines.

Limits

. The authors within the limits in addition to the specific ones should mention the generic ones given by the administration of a questionnaire.

Minor concerns

I ask the authors to expand the literature on the subject comparing their data with that reported by other studies. The literature should be greatly expanded.

Author Response

Response to Reviewer 1 Comments

Point 1:

Introduction:

Authors should acknowledge that media information has helped fuel the emergence of conspiracy theories as well as providing correct information. The administration of conflicting information has contributed to creating hesitations in a certain segment of the population. [cfr The Waiver of Patent Protections for COVID-19 Vaccines during the ongoing Pandemic and the Conspiracy Theories: Lights and Shadows of an Issue on the Ground. Front. Med. 8:756623. doi: 10.3389/fmed.2021.756623].

Another aspect that needs to be clarified is that of the fear of COVID-19. According to the authors, this aspect (considering the data of China better than other states for incidence and mortality) may have influenced the opinion of the interviewees? [cf 10.22514 / sv.2021.098].

Response 1:

Thank you for pointing this out. Based on the reviewer's comments, we have made the appropriate additions in [1.Background ], as follows:

“In addition, media information has helped drive negative press and ultimately led to vaccine hesitation [12].”

“In addition, relevant studies have shown that media trust often has cognitive and emotional effects on recipients [22]. For example, high media trust often leads to increased concern about influenza [23], while low media trust causes uncertainty to increase emotional responses such as fear and worry [24]. Previous studies have found that behavior theories can explain behavior from social and mental health perspectives and help medical and health care providers design effective interventions to alter specific behaviors [25,26]. ”

Point 2: 

Methods:

1)The authors should specify how the calculation of the sample size was made.

Response 2: As Reviewer suggestion, we have made the appropriate additions in [2.1. Participants and procedures ], as follows:

“As structural equation modeling guarantees a sample size to observed variable ratio observed Variable of at least 10:1, with 54 observed variables in this study, a minimum of 540 subjects would be required. ”

Point 3: 

Discussion:

In the discussion, the authors could evaluate whether the same type of study was carried out in other countries by comparing the results in a few lines.

Response 3: 

We have made the appropriate additions in [4.2. Role of media trust of COVID-19 vaccination], as follows:

“The findings of this study also differ from those of Nioi [12] and Erfei [45] et al. In foreign countries, different types of coverage of the same issue may occur due to the different political orientations of media outlets. Therefore public trust in different media often leads to behavioral differences such as reduced protective behavior or increased vaccine hesitancy [46]. In China, however, both national and local traditional media outlets tend to have the same political orientation, i.e., advocating for the public to be vaccinated as much as possible. Therefore, a high level of trust in traditional media helps to reduce vaccine hesitancy and increase future vaccination intentions.”

“In this study, social media trust had no effect on perceptions of COVID-19 severity and susceptibility, which is consistent with Ranjit's findings in the USA [34]. Although social media trust also directly impacted self-efficacy, response efficacy, and response costs, the degree was slightly lower than traditional media trust, related to social media participation and information diversification [47]. Reviews and widespread discussion of COVID-19 can still be found on social media such as Weibo, and misinformation or disinformation, including "conspiracy" may affect the proper understanding of the health, social and economic threats posed by COVID-19 [48] and may reduce political support [49]. ”

“A relevant study had shown that reducing harmful advice by only 10% can significantly curb risk-taking behaviors [50], so addressing the theory of conspiracy may be as important as disseminating health advice regarding public health, given the scope, speed, and scale of social media.”

Point 4: 

Limits

The authors within the limits in addition to the specific ones should mention the generic ones given by the administration of a questionnaire.

Response 4: Based on the reviewer's comments, we have made the appropriate additions in [4.3. Limitations], as follows:

“Thirdly, using an online survey may limit the representativeness of the results, so this study used a large sample size and random stratified sampling to address this limitation. ”

“In addition, this study published the questionnaire in an electronic format, thus excluding those who could not use a smartphone to answer the questionnaire, particularly those who were older, and therefore the study lacked representation of those aged 50 and over.”

Point 5: 

Minor concerns

I ask the authors to expand the literature on the subject comparing their data with that reported by other studies. The literature should be greatly expanded.

Response 5: As the Reviewer suggested that expanding the literature on the subject, we have made the appropriate additions. However, due to the limited literature related to media trust and vaccination, relevant literature was mainly cited for comparison and discussion in [4. Discussion], while appropriate additions are made in [1.background] section.

Reviewer 2 Report

An interesting manuscript analyzing the impact of media trust in people's motivation to receive COVID-19 vaccination using the protection motivation theory. Definitely, the manuscript is within the scope of the journal and very interesting since COVID-19 is still a global problem!

The manuscript is well written and structured, there are only minor comments:

  1. The authors mentioned that “Pearson linear correlation coefficient matrix was used to explore the relationships between variables”, however Pearson correlation coefficient is used when there is normality in the variables, thus it is important to mention in “2.3. Data analysis” section which test was applied to ensure data normality and what were the results. Perhaps is better to apply instead of Pearson correlation a Spearman test. Really do not think that the conclusions will e different.
  2. Table 4 is extremely difficult to read perhaps is better to use old entries when p<0.05 and bold+italics entries when p<0.0001
  3. This study was conducted electronically (albeit through the mobile phone) however, people that find it difficult to use the mobile phone interface to answer the questionnaire are excluded (especially higher ages). Thus I would propose to add another study limitation in the discussion. This is also reflected by the small percentage of participants aged >50 (3.34% as depicted in table 1), while China's population in a very smaller group 50-70 years old is 19.49% as is already mentioned in the manuscript. In reality to have a fair representation of the age groups, one would expect higher than 20% of participants in the age group >50.

Author Response

Response to Reviewer 2 Comments

Point 1: The authors mentioned that “Pearson linear correlation coefficient matrix was used to explore the relationships between variables”, however Pearson correlation coefficient is used when there is normality in the variables, thus it is important to mention in “2.3. Data analysis” section which test was applied to ensure data normality and what were the results. Perhaps is better to apply instead of Pearson correlation a Spearman test. Really do not think that the conclusions will e different.

Response 1: We have re-considered and made correction in [2.3. Data analysis] according to the Reviewer’s comment. As we know that compared to the Pearson correlation, Spearman does not require the distribution of the original variables, it is more widely used, and the results obtained from the data are not significantly different from the previous, but more appropriate, so the correlation analysis was adjusted for this study, as detailed in Table 4.

Point 2: Table 4 is extremely difficult to read perhaps is better to use old entries when p<0.05 and bold+italics entries when p<0.0001.

Response 2: We have corrected Table 4 of [3. Results] according to the Reviewer’s comment, which made the it more easier to read.

Point 3: 

This study was conducted electronically (albeit through the mobile phone) however, people that find it difficult to use the mobile phone interface to answer the questionnaire are excluded (especially higher ages). Thus I would propose to add another study limitation in the discussion. This is also reflected by the small percentage of participants aged >50 (3.34% as depicted in table 1), while China's population in a very smaller group 50-70 years old is 19.49% as is already mentioned in the manuscript. In reality to have a fair representation of the age groups, one would expect higher than 20% of participants in the age group >50.

Response 3: Based on the reviewer's comments, we have made the appropriate additions in [4.3. Limitations], as follows:

“In addition, this study published the questionnaire in an electronic format, thus excluding those who could not use a smartphone to answer the questionnaire, especially those in older grades, and therefore, there were fewer participants aged 50 and above in this study.”

Reviewer 3 Report

Thanks for inviting me to peer review the study titled "Analysis of the impact of media trust on the public’s motivation to receive future vaccinations for COVID-19 based on protection motivation theory", by Dr Li and Dr Sun. Through an extensive survey conducted in China, they explore the impact of traditional media, social media and interpersonal communication on vaccination hesitancy.

The study presents some interesting findings. It would benefit from better presentation of the statistical methods used (especially around the construct of the SEM model). The SEM model should be explained for the less experienced reader and should describe how is potential confounding accounted for (in methods & results). Also, authors should clarify which of the findings result from analyses that accounted for confounding vs not.

Moreover, the pandemic has been covered differenently by media in different countries. In some countries there is a lot of debate around vaccines/ treatments on traditional and social media, while other countries cultivated consensus and strictly disseminated evidence-based data (pro-vaccination). It would be beneficial for the reader to have some information around the media approach in China

Author Response

Response to Reviewer 3 Comments

Point 1: The study presents some interesting findings. It would benefit from better presentation of the statistical methods used (especially around the construct of the SEM model). The SEM model should be explained for the less experienced reader and should describe how is potential confounding accounted for (in methods & results). Also, authors should clarify which of the findings result from analyses that accounted for confounding vs not.

Response 1: Based on the reviewer's comments, we have made the appropriate additions in [2.3. Data analysis], as follows:

“SEM is a multivariate statistical method for modeling relationships utilizing covariance of variables. Generally speaking, SEM can be decomposed into two parts: measurement model and structural model. The former refers to the relationship between indicators and latent variables and deals mainly with the measurement of latent variables; the latter deals with the relationship between latent variables and with observed variables other than indicators measured by non-latent variables and deals mainly with the causal relationships between different concepts. Hence SEM is often used to test the relationships between theoretical hypotheses. In this study, we will use SEM to explore the relationship between different channels of media trust (observed variables) and each of the latent variables of PMT and assesses the impact on vaccination intention. ”

Point 2: Moreover, the pandemic has been covered differenently by media in different countries. In some countries there is a lot of debate around vaccines/ treatments on traditional and social media, while other countries cultivated consensus and strictly disseminated evidence-based data (pro-vaccination). It would be beneficial for the reader to have some information around the media approach in China.

Response 2: We have made correction according to the reviewer’s comment in [4.2. Role of media trust of COVID-19 vaccination], as follows:

“During the COVID-19 vaccination period, the Chinese government provided free COVID-19 vaccination services to the general public, advocating that the public receive the COVID-19 vaccine "as much as possible if they should" to improve herd immunity. Government media such as national and local television news and newspapers promoted vaccination regarding indications, safety, vaccination procedures, and free vaccination. At the same time, medical experts publicized it from newly confirmed cases, mass vaccination, virus variability, and vaccine protection effect, so that people with a high level of trust in traditional media not only had less reason to hesitate about the vaccine (support for NPI and worrying about safety and efficacy) but also were motivated to take the protective measures recommended by the government. ”

“The findings of this study also differ from those of Nioi [12] and Erfei [45] et al. In foreign countries, different types of coverage of the same issue may occur due to the different political orientations of media outlets. Therefore public trust in different media often leads to behavioral differences such as reduced protective behavior or increased vaccine hesitancy [46]. In China, however, both national and local traditional media outlets tend to have the same political orientation, i.e., advocating for the public to be vaccinated as much as possible. Therefore, a high level of trust in traditional media helps to reduce vaccine hesitancy and increase future vaccination intentions.”

“In this study, social media trust had no effect on perceptions of COVID-19 severity and susceptibility, which is consistent with Ranjit's findings in the USA [34]. Although social media trust also directly impacted self-efficacy, response efficacy, and response costs, the degree was slightly lower than traditional media trust, related to social media participation and information diversification [47]. Reviews and widespread discussion of COVID-19 can still be found on social media such as Weibo, and misinformation or disinformation, including "conspiracy" may affect the proper understanding of the health, social and economic threats posed by COVID-19 [48] and may reduce political support [49]. As a result, individuals with high trust in social media did not have much motivation to comply with measures recommended by the government. ”

Special thanks to you for your good comments.

Round 2

Reviewer 1 Report

I thank the authors for the corrections made. I find the paper has improved significantly. I have no further observation to make.